

# Review of materials and testing methods for virus filtering performance of face mask and respirator

Bhanu Bhakta Neupane[1] and Basant Giri[2]

[1] Central Department of Chemistry, Tribhuvan University, Kathmandu, Nepal
[2] Center for Analytical Sciences, Kathmandu Institute of Applied Sciences, Kathmandu, Nepal

## ABSTRACT

Respiratory protection devices such as face masks and respirators minimize the transmission of infectious diseases by providing a physical barrier to respiratory virus particles. The level of protection from a face mask and respirator depends on the nature of filter material, the size of infectious particle, breathing and environmental conditions, facial seal, and user compliance. The ongoing COVID-19 pandemic has resulted in the global shortage of surgical face mask and respirator. In such a situation, significant global populations have either reused the single-use face mask and respirator or used a substandard face mask fabricated from locally available materials. At the same time, researchers are actively exploring filter materials having novel functionalities such as antimicrobial, enhanced charge holding, and heat regulating properties to design potentially better face mask. In this work, we reviewed research papers and guidelines published primarily in last decade focusing on, (a) virus filtering efficiency, (b) impact of type of filter material on filtering efficiency, (c) emerging technologies in mask design, and (d) decontamination approaches. Finally, we provide future prospective on the need of novel filter materials and improved design.

## INTRODUCTION

We have witnessed several viral disease outbreaks in recent decades. Few notable examples of such outbreak include severe acute respiratory syndrome coronavirus-2 (SARS-CoV-2) in 2019 (*Chen et al., 2020*), the Ebola virus in 2014 (*World Health Organization, 2014*), Middle East respiratory syndrome coronavirus (MERS-CoV) in 2012 (*Assiri et al., 2013*), the influenza pandemic (H1N1) in 2009 (*Yang et al., 2009*) and severe acute respiratory syndrome coronavirus-1 (SARS-CoV-1) in 2002-2003 (*Donnelly et al., 2003*). Respiratory infection spreads through surface contact, droplet spray, and airborne modes of transmission (*Atkinson & Wein, 2008*; *Cowling et al., 2013*; *Wei & Li, 2016*). However, the relative contribution of each mode of transmission is not completely understood for many viruses (*Janssen et al., 2013*). The recent evidences have shown that COVID-19 is transmitted through contact, respiratory droplets (*Huang et al., 2020*; *Peeri et al., 2020*;

Corresponding author
Bhanu Bhakta Neupane,
bbneupane@cdctu.edu.np

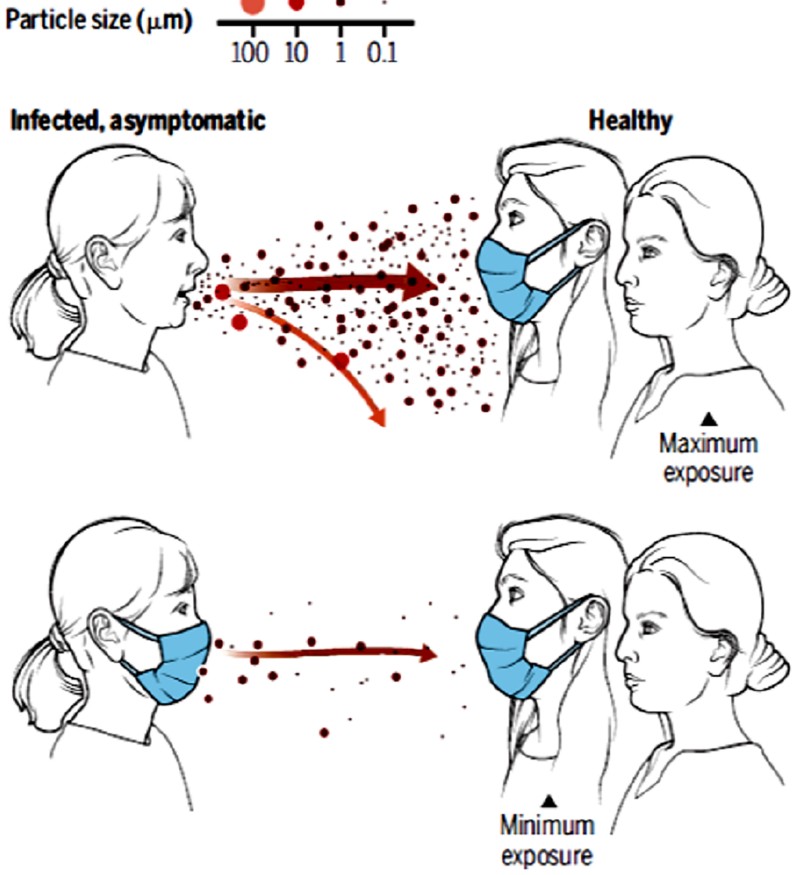

**Figure 1 A cartoon showing the importance of face mask in reducing transmission.** Reproduced with permission from *Prather, Wang & Schooley (2020)*. Copyright © 2020, American Association for the Advancement of Science.

*World Health Organization, 2020*), and aerosol routes (*Greenhalgh et al., 2021*). The various mode of transmission can be partly or fully interrupted by a combination of personal hygiene practices such hand washing, use of personal protective equipment (PPE) such as face mask and respirator, and physical distancing (*Siegel et al., 2007*).

Viruses containing muco-salivary droplets (>5 µm) and aerosol particles (≤5 µm) are exhaled from an infected individual (both symptomatic and asymptomatic) during speaking, breathing, coughing, and sneezing activities. These particles can travel few meters in air depending on their size, gravitational settling, and evaporation rate (*Yang et al., 2007a*; *Prather, Wang & Schooley, 2020*). Face masks and respirators significantly minimize and/or prevent the spread of infection by creating a physical barrier to the virus particles and droplets (*Rengasamy, Zhuang & Berryann, 2004*; *MacIntyre & Chughtai, 2015*). Particle exposure is maximum if physical distance is less than six feet and mask is not worn (Fig. 1). Alternatively, maximum protection is achieved if both healthy and infected individuals properly use a recommended mask and physical distancing is maintained (*World Health Organization, 2020*).
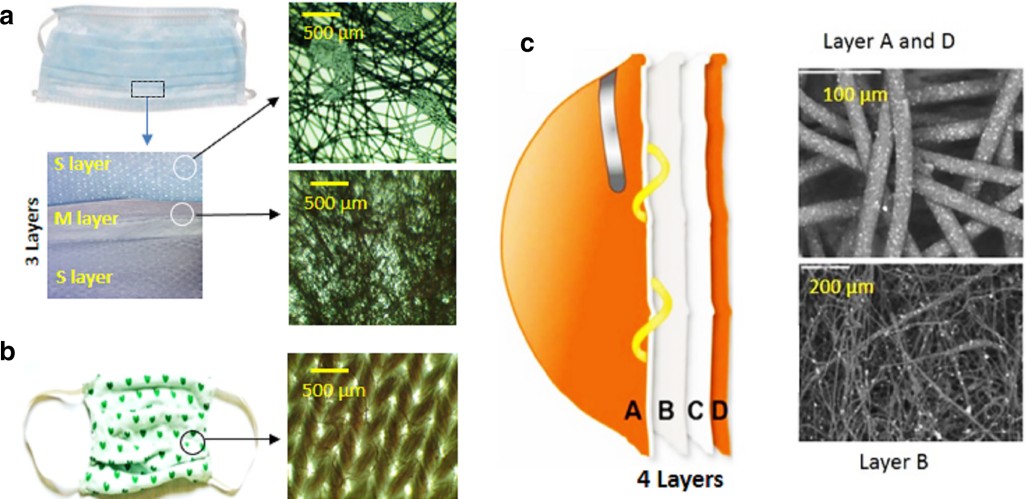

**Figure 2 Material used in face mask and respirator.** (A) SMS type surgical mask and optical microscopic images of the outermost and innermost layers (the S layers) and middle layer (the M layer). (B) A typical cloth face mask and optical microscopic image of the cloth face mask surface. Reproduced with permission from *Neupane et al. (2019)*. (C) Multilayered structure of a respirator. Letters A and D represent the spun bonded polypropylene layers, B the melt blown layer, and D the support layer. The scanning electron microscopic (SEM) image layers A and D and B are also shown. Reproduced with permission from *Borkow et al. (2010)*.

## Surgical face mask

A properly worn surgical mask creates a physical barrier between the immediate environment and respiratory orifices (mouth and nose) thereby blocking or minimizing in and out movement of infectious droplets and particles (*Seale et al., 2009*). These masks are labeled as isolation, dental, or medical procedure masks and they are generally known as facemasks. However, all facemasks are not regulated as surgical masks. In USA, the Food and Drug Administration (FDA) regulates surgical masks under 21 CFR 878.4040. The FDA regulation requires that the surgical mask must have recommended filtering efficiency for inert particles and biological particles, fluid barrier protection standards and flammability tests (see "Filtering performance of face masks and respirators") (*US Food and Drug Administration, 2021*). Surgical masks are designed for one time use and they generally do not provide good facial seal. The US FDA approved surgical facemasks are recommended for general public and health care professionals at medium to low risk settings (*Seale et al., 2009*). Three layered flat or cup shaped masks with stretchable ear loops or straps are the most commonly used surgical masks. Each layer in the three-layered surgical mask is designed to have a unique functionality (Fig. 2A). Details on virus filtering performance and material design of surgical mask is provided in later sections.

There has been a shortage of surgical masks during viral outbreaks including the ongoing COVID-19 due to high demand in the global market (*Wu et al., 2020*). In such difficult situations, general public wear homemade or locally made cloth face mask (Fig. 2B) and or face covering. However, homemade masks provide limited protection to the user (*Chughtai, Seale & MacIntyre, 2013*; *MacIntyre et al., 2015*).

## Respirators and their classification

Respirators are recommended for high risk environments; for example for health professionals when providing care to COVID-19 patients (*Seale et al., 2009*; *Phan et al., 2019*; *World Health Organization, 2020*). The outer rim of respirator provides better seal or fit around the nose and mouth. Since respirators have special design and advanced filter materials (Fig. 2C), a properly worn respirator provides an expected protection to the user. Details of filtering performance and material design of a respirator is provided in later sections.

A proper respirator must be selected for a specific hazard. In USA, filtering facepiece respirators (FFRs) (disposable half-facepiece respirators), elastomeric half-and full-facepiece respirators, powered air-purifying respirators, supplied air respirators, self-contained breathing apparatus and combination respirators are available. The FFRs are available in N, P and R series having minimum filtering efficiency of 95 (N95, P95 and R95), 99 (N99, P99 and R99) and 99.97% (N100, P100 and R100) respectively for particles having aerodynamic mass median diameter of 0.3 μm. The letters N, P, and R refer to resistant to oil, partially resistant, and resistant (oil proof); respectively (*OSHA, 1996*). Other countries have different types of respirators. In UK, FFRs are available as FFP1, FFP2, and FFP3 having minimum filtering efficiency of 80%, 94% and 99.95%; respectively.

The N95 FFRs have reasonable cost and are widely used. Surgical N95 respirator are a subset of N95 FFRs which are used in healthcare settings. In USA, surgical N95 respirators are NIOSH certified under 42 CFR Part 84 (*US FDA, 1995*). NIOSH reviews the results for fluid resistance, flammability, pressure difference, and biocompatibility supplied by the manufacturer for certification. FDA also provides clearance to surgical N95 (under 21 CFR 878.4040) for fluid resistance, flammability, and biocompatibility properties. Commercial respirators come with or without exhalation valve (EV). The EV helps to minimize excessive dampness and heating and it offers decreased breathing resistance to the user. However, a faulty EV could contaminate the nearby environment with infectious virus particles thereby decreasing the level of protection. Surgical N95 respirator without EV or equivalent is recommend for MERS-CoV, SARS-CoV-1 and SARS-CoV-2 for health care professionals in high risk environments (*CDC-NIOSH, 2020*).

The filtering efficiency of facemask and respirator for neutral NaCl aerosol and other particles are well documented in multiple studies (*Rengasamy et al., 2017*; *Sharma, Mishra & Mudgal, 2020*; *Palmieri et al., 2021*) but the information on virus filtering efficiency is not well described. Also, during the COVID-19 pandemic virus filtering efficiency of facemask and respirator can be of special interest to readers. This review provides comprehensive description of filter media used in face masks and respirators, their virus filtering efficiency, emerging technologies for better performing masks and respirators, and decontamination approaches. We also provide future prospective on the need of novel filter material and improved design.

## SURVEY METHODOLOGY

We used Google Scholar and PubMed platforms to search relevant documents published before February 2021. The keyword used were "facemask and virus filtering efficiency" OR "facemask and virus" OR "facemask and material". We reviewed abstract of the documents and selected documents that provided new insight on the virus filtering performance of either cloth facemask, surgical facemask, or respirators were considered further. Additionally, documents that reported significant advancement in the material design and or the understanding the facemask filter materials were also included. Patents were excluded in the study.

## FILTERING PERFORMANCE OF FACE MASK AND RESPIRATOR

### Filtering efficiency

A filtering device, here referring to both face mask and respirator, provides a barrier protection to the user by capturing infectious droplets, bio-aerosol, and other particles. Conventional single fiber filtration theory (*Raist, 1987*) predicts that the particles bigger than 0.3 μm are captured on the filter mainly by interception and inertial impaction and particles smaller than 0.2 μm are captured by diffusion and electrostatic attraction or polarization effects. None of the capture mechanisms are dominant for intermediate sized particles (0.2–0.3 μm), which are known as the most penetrating particles size (MPPS) (*Hinds, 1999*; *Hakobyan, 2015*). MPPS depends on the nature of filter material and ranges from 0.03 to 0.1 μm (*Shaffer & Rengasamy, 2009*).

Filtering efficiency of a device depends on multiple parameters such as property of material used in the device, size of particle, and environmental factors. Filtering efficiency (*E*) is one the most important parameters to quantify filtering performance of face mask and respirator and is given by Eq. (1),

$$E = \left(1 - \frac{C_i}{C_o}\right) \times 100 \tag{1}$$

where, $C_i$ and $C_o$ are the concentration of particles inside (downstream) and outside (upstream) the filtering device. Alternatively, the performance is also measured in terms of the penetration efficiency (*P*) (*Johnston et al., 1992*);

$$P = 100 - E = \left(\frac{C_i}{C_o}\right) \times 100 \tag{2}$$

The overall filtering performance of a respirator or class of respirators is also measured in terms of assigned protection factor (APF). APF is defined as the level of respiratory protection that a respirator or class of respirators is expected to provide to the user at the workplace when the employer implements an effective respiratory protection program on continuous basis as specified in the 29 CFR 1910.134 standard (*OSHA, 2009*). AFP of a respirator cannot be measured simply from the known aerosol particles inside ($C_i$) and outside ($C_o$) the respirator. Several factors such as nature of filter media, length of exposure, facial seal, nature and concentration of contaminants, duration of exposure are

considered while assigning the APF (*Janssen et al., 2013*). APF of 10 and 20 means a respirator reduces the exposure level by a factor of 10 and 20, respectively.

### *In vitro* measurement of filtering efficiency

The filtering efficiency of a filtering device varies with test parameters used such as particle/aerosol size and distribution, filter media charge and particle charge, face velocity, humidity, and flow rate. For example, the filtering efficiency decreases with increase in relative humidity, face velocity, and flow rate (*Yang et al., 2007b*; *Thakur, Das & Das, 2013*). The regulatory recommendations are made from *in vitro* measurement of filtering efficiency (*NIOSH, 1996*). To ensure that a device can filter even the most penetrating particles in a workplace, the filtering efficiency is measured at standard testing conditions that mimic the *worst-case scenario* in the workplace.

NIOSH uses NaCl aerosol method for testing N95 FFRs (*NIOSH, 2019*). The test parameters are:

a) flow rate of 85 L min$^{-1}$ that simulates the breathing volume during a heavy work load,

b) poly-dispersed and charge neutralized sodium chloride aerosol particles having count median diameter (CMD) of 75 ± 20 nm and geometric standard deviation (GSD) of <1.86 or broad range distribution (log-normal distribution) NaCl aerosol particles having mass median aerodynamic diameter (MMAD) and mass median diameter (MMD) about 300, 240 nm, respectively (*Bollinger, 2004*; *Shaffer & Rengasamy, 2009*),

c) aerosol particle concentration of <200 mg/m$^3$,

d) pre-conditioning of the filtering device at ~85% relative humidity and ~38 °C for 24 h.

Different test methods are used for testing the performance of material used in surgical face mask. In the ASTM F2229-03 method, surgical face mask material is challenged with charge neutralized latex spheres having size range of 0.1-5 µm at airflow test velocities (face velocity) of 0.5 to 25 cm/s (*ASTM, 2003*). Recommended aerosol concentration is $10^7$-$10^8$ particles/m$^3$ and can be diluted if needed. The US FDA recommends slightly different test parameters for testing material (not the entire mask) used in surgical face mask. This method recommends 0.1 µm charge un-neutralized polystyrene latex spheres at the air flow velocity of 0.5 to 25 cm/s (*US Food & Drug Administration, 2004*). The bacterial filtering efficiency (BFE) is measured as per FDA guidance and ASTM F2101 method (*ASTM, 2001*). The mask material (not the whole mask) is challenged with un-neutralized *S. aureus* bacterial aerosol (mean particle size of 3 ± 0.3 µm diameter) at a flow rate of 28.3 L/min. The US FDA provides clearance to surgical mask after reviewing the information provided by the manufactures in the 510(k) premarket application (*US Food & Drug Administration, 2004*). In the 510(k) application, manufactures are required to provide data of fluid resistance, polystyrene latex and *S. aureus* bacterial aerosol filtering efficiency, differential pressure, and flammability tests.

## Virus filtering efficiency of surgical masks and respirators

Performance of filtering device can also be measured using virus aerosol particles to calculate virus filtering efficiency (VFE). *Eninger et al. (2008)* measured the filtering efficiency of N95 and N99 respirators using three different virus aerosols (enterobacteriophages MS2, T4, and *Bacillus subtilis* bacteriophage) and NaCl aerosol particles at three different inhalation flow rates, 30, 80, and 150 L min$^{-1}$. The filtering efficiency of both N95 and N99 respirator was ≥96% for the 0.02–0.5 μm of aerosol particles. Similar filtering efficiency was reported for virus aerosols suggesting that neutral NaCl aerosols may be appropriate for mimicking the filter penetration of similar size viruses. It is to be noted that the virus efficiency test is not required by FDA or NIOSH for approval process.

*Balazy et al. (2006)* measured the VFE of NIOSH certified N95 respirators and surgical masks using MS2 virus. In the study, VFE of >95% was achieved using virus aerosol particles (10–80 nm) and inhalation flow rate of 85 L/min. For the test conditions, VFE of two types of surgical masks was found to be ~80% and ~15%, suggesting surgical masks cannot be as effective as N95 respirators for small virus.

*Shimasaki et al. (2018)* measured the penetration efficiency of two types of nonwoven fabrics SMS type and S (Spunlace) that are used in commercial surgical masks using ΦX174 phase and inactivated influenza virus aerosols at a flow rate of 15 L min$^{-1}$. The hydrodynamic diameter of the phase and influenza virus as determined by dynamic light scattering was 28 and 112 nm, respectively. The penetration efficiency for ΦX174 phase and influenza virus was ~6% and 20% for SMS type and ~30% and 80% for S type, respectively. The three layered in SMS type may have provided lower penetration efficiency or higher efficiency. In another study, filtering efficiency of surgical N95 respirator was ≥99.6% for all combinations of experiment configurations using influenza A virus, rhinovirus 14, and bacteriophage ΦX174 at a flow rate of 28.3 L min$^{-1}$ (*Zhou et al., 2018*).

Even though it is challenging to study the filtering efficiency of masks using *viable* virus aerosol particles, few studies have reported such experiments. *Harnish et al. (2013)* measured the VFE of NIOSH approved N95 respirators using viable H1N1 virus aerosolized in artificial saliva buffer (CMD of 0.83 μm) at a flow rates of 85 and 170 L min$^{-1}$. The respirator was glue sealed in a six inch diameter sample holder. The N95 respirator provided VFR of 99.3% at both flow rates. They also measured the filtering efficiency using 0.8 μm polystyrene latex beads aerosol and got similar filtering efficiency. This study suggested that the dead or live status of aerosol does not affect filtering efficiency. In another study (*Harnish et al., 2016*), VFE of five different models of NIOSH certified N95 respirators was measured using viable H1N1 influenza aerosol and polystyrene latex bead aerosols having CMD of 0.1 μm representing MPPS for commonly used filter media at a flow rate of 85 L min$^{-1}$. Mean VFE of respirators sealed to the sample holder ranged from 99.23% to 99.997% and particle aerosol filtering efficiency ranged from 99.17% to 99.995%. This study suggested that the N95 respirators can be used for protection against H1N1 virus in workplace. They also confirmed the earlier conclusion

**Table 1 Filtering efficiency.**

| Filtering device | Filtration efficiency | Major test parameters |
|---|---|---|
| N95 | ≥95% | MS2 virus aerosol, flow rate 85 L min$^{-1}$ (*Balazy et al., 2006*) |
| | ≥99.2% | H1N1 viable virus, flow rate 85 L min$^{-1}$ (*Harnish et al., 2016*) |
| | 97.1–97.8% | bacteriophage phiX174, 28.3 L min$^{-1}$ (*Rengasamy et al., 2017*) |
| Three layered surgical mask | ~85% | MS2 virus aerosol, flow rate 85 L min$^{-1}$ (*Balazy et al., 2006*) |
| | ~94% | bacteriophage phiX174, flow rate 15 L min$^{-1}$ (*Shimasaki et al., 2018*) |
| | ~80% | Influenza virus, flow rate 15 L min$^{-1}$ (*Shimasaki et al., 2018*) |
| | ~90% | bacteriophase MS2, flow rate 30 L min$^{-1}$ |
| Two layered cloth mask | 50–70% | bacteriophase MS2, flow rate 30 L min$^{-1}$ |

(*Harnish et al., 2013*) that the dead or live status of aerosol does not affect the filtering efficiency of a respirator.

## Virus filtering efficiency of cloth face masks

The particulate matter filtering performance of cloth mask has been found to be lower than commercially available surgical masks and respirators (*Rengasamy, Eimer & Shaffer, 2010*; *Shakya et al., 2017*; *Neupane et al., 2019*). *Davies et al. (2013)* reported the filtering efficiency of two layered cloth mask made from commonly available fabrics using bacteriophase MS2 virus aerosol (~23 nm diameter) at a flow rate of 30 L min$^{-1}$. The percentage filtering efficiency of cloth masks made of 100% cotton, scarf, tea towel, pillowcase, cotton mix, linen, and silk were 50.85 ± 16.81, 48.87 ± 19.77, 72.46 ± 22.60, 57.13 ± 10.55, 70.24 ± 0.08, 61.67 ± 2.41, 54.32 ± 29.49, respectively. The filtering efficiency of three ply surgical mask was better (89.52 ± 2.65%). This study suggested that homemade mask should be considered as the last option. Such masks should be worn only if no better mask is available.

A cluster randomized trial of cloth masks in healthcare workers in hospital settings reported that influenza like illness was higher in healthcare workers who wore cloth mask than those who wore surgical mask (*MacIntyre & Chughtai, 2015*). In a recent study (*Leung et al., 2020*), a significantly lower amount of coronavirus RNA in respiratory droplet and aerosols and influenza virus RNA in respiratory droplets was found in patients who wore surgical masks than in patients who did not wear surgical masks. This study suggested that surgical face masks could be used by COVID-19 patients to reduce onward transmission.

A summary of virus filtration efficiency of face masks and respirators along with few major test parameters is summarized in Table 1.

## FILTERING EFFICIENCY AND MATERIAL PROPERTY

An important question one can have at this point is: *What makes the filtering efficiency different*? Provided the same test parameters used, the observed difference in filtering efficiency of face masks and respirators is due to: (a) inherent property of the filter

material, and (b) facial fitness *i.e.*, how well the filtering device fits onto the face, and (c) breathing condition.

A key component of commercially available surgical masks and respirators is a non-woven filter membrane. The membrane consists of 1–20 μm diameter fibers oriented randomly. The material can be fabricated from synthetic or natural polymers or composites such as polypropylene and polyethylene by melt blowing technique. The membrane is mostly electrostatically charged and called as *electret filter*. The charge is imparted onto the membrane by corona discharge, induction charging, and tribo-electric techniques during fabrication (*Thakur, Das & Das, 2013*; *Hutten, 2015*). In contrast to conventional filter, electret filter provides better particle capture efficiently by electrostatic interaction. Also, the downstream air pressure drop in such filter is lower resulting in lower resistance to breathing which is referred to as better breathability (*Thakur, Das & Das, 2013*; *Zhang et al., 2018*).

The filtering efficiency of electret filter depends on charge density, charge retaining or holding capacity, and size (length and diameter) and arrangement of fiber. These parameters depend on the material type and filter manufacturing technique to a great extent. It known that filter having smaller fiber diameter leads to higher filtration efficiency than the fiber having lager diameter, but the pressure drop in the former is higher making the filtering device less breathable. In addition, shorter fiber form more porous filter than longer fiber. Filtering efficiency increases with increase in filter thickness at the expense of breathing resistance. The columbic and di-electrophoretic forces are also known to be stronger in filter having smaller fiber diameter. This results in stronger capturing of pathogens and better protection. Spun bonding and melt blowing are the most commonly adopted techniques for the fabrication of fibrous filter membrane. The melt blowing technique produces filters having smaller fiber diameter. Therefore, this is the method of choice in manufacturing of filtering media used in surgical mask and respirators (*Thakur, Das & Das, 2013*).

The charge density on the electret media affects the filtering performance. Charge intensity and storage capacity depends on the dielectric property of a fiber material. In general, the polymeric materials having high electrical resistance, thermal stability, and hydrophobicity (for example; polypropylene, polyethylene) provide better charge storage ability and stability (*Van Turnhout, Adamse & Hoeneveld, 1980*). If charge on the electret media is removed, then the filtering efficiency decreases significantly. The *most penetrating particle size* (MPPS) for electret media varies with material properties including charge density and is reported in the range of 0.03–0.1 μm (*Hinds, 1999*; *Shaffer & Rengasamy, 2009*; *Hakobyan, 2015*). That is why the filtering performance of a filtering device is measured by using particles having size at or close to MPPS.

## Material design of a surgical mask

In the mostly commonly used three layered/ply SMS type surgical mask, for example US FDA approved surgical face mask, the middle layer fabricated by melt blown technique (the M layer) is sandwiched between outer and inner layers fabricated by spun bonded technique (the S layers). The three layers are designed to have specific functions.

In all layers, fibers are randomly oriented (non-woven) so as to form web like arrangement (Fig. 2A and inset). The fiber density in middle layer is higher than in the other two layers resulting low porosity. Since this layer is charged, it can efficiently capture infectious particles above. The outermost layer (typically coded blue) is hydrophobic and limits the penetration of water rich muco-salivary droplets. The innermost layer is hydrophilic and can absorb spit, sweat, and muco-salivary droplets thereby minimizing dampness and increasing user comfort. In recent years, surgical face mask having additional functionality are also being explored (see the "Emerging Technologies" section).

## Material design of a cloth mask

For comparison, we also like to comment on the material property of cloth or fabric face masks. The cloth masks are made from woven or knitted fabrics and are mostly two layered. Commonly used cloth face masks have variable pore size and thread density depending on the nature of the fabrics (Fig. 2B and inset). The lower performance of cloth facemask (*Rengasamy, Eimer & Shaffer, 2010*; *Shakya et al., 2017*; *Neupane et al., 2019*; *Konda et al., 2020*) is also due to poor facial fitness and performance of material used. Studies have suggested to use tightly woven fabrics having high thread count and low porosity, such as quilting cotton and cotton sheets, to design relatively better performing cloth face masks (*Konda et al., 2020*; *Neupane, Chaudhary & Sharma, 2020*). The efficiency can be further increased by increasing the number of fabric layers. However, more fabric layers increase the breathing resistance making the mask uncomfortable for use (*Drewnick et al., 2020*; *Hancock et al., 2020*; *Konda et al., 2020*; *Zangmeister et al., 2020*).

## Material design of a N95 respirator

A typical NIOSH certified N95 respirator consists of four layered structure (labeled A, B, C, D in Fig. 2C). The outer most layer A (farthest from the face) and D (closest form the face) are made from spun bonded polypropylene (*Borkow et al., 2010*). These layers contain larger sized fibers and capture coarse particles and stop moisture entering into the inner layers. The inner layer B is made from melt blown polypropylene. It is charged (electret membrane) and contains highly packed small fibers (*i.e.*, low porosity) and eventually can filter fine particles. The next inner layer C is made from a plain polyester and gives a shape to the respirator. This gradient filtration mechanism in the respirator provides high filtering efficiency. Another important factor for better performance of respirator is its design that provides excellent facial fit. Surgical masks are not designed to fit tightly on the face, so they cannot provide the same level of protection as the respirators (*CDC-NIOSH, 2020*). It is to be noted that the NIOSH certification does not look at the number of filter media layers and the order of hydrophobic and hydrophilic layers. But the N95 respirators should meet the standard test requirements as described in the 42 CFR Part 84.

## EMERGING TECHNOLOGIES

Several efforts have been reported to make better performing masks and respirators in past. Such efforts involve technologies for making new or modified filter pieces, manufacturing

protocols, and disinfecting procedures among others. To address the shortage of standard face mask and respirator in emergency situation, researchers, manufacturers, local hospitals, and even general public have proposed a number of innovative ideas.

## 3D printing of mask accessories

Additive manufacturing (AM) including 3-dimensional (3D) printing have gained popularity in manufacturing medical devices (*Ventola, 2014*). 3D printing has been used to make mask components such as mask structure or frame, cover, filter fix, seal etc. A variety of different types of materials including polymax PLA filament, SLS/MJF nylon or flexible SLA resin have been used. Foam or silicone band have been used to print seal with improved airtightness and softer skin touch. Even though the 3D printed masks may look like conventional PPE, they may not provide the same level of barrier protection, fluid resistance, filtration, and infection control (*Ventola, 2014*; *Morrison et al., 2015*). The new designs are not approved by any regulatory agencies yet and performance may have been compromised. Since the 3D printed masks and accessories provide low-cost, quick, and decentralized and distributed manufacturing, they can be promising alternatives during emergency situation. The US FDA has developed preliminary guidance to devices using AM that involves 3D printing (*Morrison et al., 2015*; *Di Prima et al., 2016*).

## Modified face mask filter media

There are several efforts to modify mask filters with various materials such as antibody and nanomaterials to enhance the antimicrobial activity and filtering efficiency of the masks. *Kamiyama et al. (2011)* reported a modified nonwoven fabric-based air filters that were impregnated with antibody for avian influenza H5N1 virus. The filters were found to inactivate the virus trapped in the filter due to antigen–antibody interaction. However, these filters were tested only for birds. All birds housed in antibody filter covered boxes did not die. Similar antibody impregnated filter could be tested for face masks. Such methods may require further research to find out how the antibody impregnated on filters would retain their activity in ambient environmental condition during transportation, storage and use of the filter. The performance of such filters while they are used in mask is not known.

Metal oxide and metal nanoparticles display biocidal activities (*Vincent, Hartemann & Engels-Deutsch, 2016*; *Fernando, Gunasekara & Holton, 2018*). In particular, copper oxide nanoparticle display potent biocidal properties against a range of microbes including bacteriophages, bronchitis virus, poliovirus, herpes simplex virus, human immunodeficiency virus and influenza viruses (*Ingle, Duran & Rai, 2014*; *Vincent, Hartemann & Engels-Deutsch, 2016*; *Fernando, Gunasekara & Holton, 2018*). Taking the advantage of the biocidal properties, respiratory face masks containing these materials have been tested for anti-microbial activities. The use of biocidal masks may significantly reduce the risk of hand or environmental contamination. They reduce infection due to improper handling and disposal of masks.

A copper oxide impregnated respiratory face mask was reported by *Borkow et al. (2010)* that demonstrated potent anti-influenza biocidal properties without altering physical

barrier properties of the masks. The copper oxide impregnation did not alter the filtering efficiency of N95 masks when tested with aerosolized viruses of human influenza A virus (H1N1) and avian influenza virus (H9N2) under simulated breathing conditions. In these experiments, no infectious H1N1 viral titers were recovered from the copper oxide containing masks within 30 min. In case of H9N2 virus, titers were recovered from the copper oxide containing masks but were five-fold lower than the control masks. The copper oxide containing masks successfully passed bacterial filtration efficacy, differential pressure, latex particle challenge, and resistance to penetration (*Borkow et al., 2010*). The metal oxide or nanoparticle impregnated respirator could have four layers of fabric as reported by *Borkow et al. (2010)*. Out of the four layers (Fig. 2c), outer two and inner layers were metal oxide impregnated polypropylene fabric and the remaining layer was made of plain polyester to give shape to the mask.

A mixture of silver nitrate and titanium dioxide nanoparticles coated facemasks were also tested against infectious agents (*Li et al., 2006*). The minimum inhibitory concentration of the nanoparticles against *Escherichia coli* and *Staphylococcus aureus* were 1/128 and 1/512, respectively. A 100% reduction in viable *E. coli* and *S. aureus* was observed in the coated mask materials after 48 h of incubation. Skin irritation was not observed in any of the volunteers who wore the facemasks.

The efficacy of four antimicrobial respirators to decontaminate MS2 virus was evaluated (*Rengasamy, Fisher & Shaffer, 2010*) using MS2 as a surrogate for pathogenic viruses. The MS2 activity of masks with antimicrobial material was significantly reduced when stored at 37 °C and 80% RH for 4 h than the masks without antimicrobial materials. The antimicrobial materials used in this research included coating of outer layer of mask with silver–copper material, incorporating EnvizO$_3$-Shield on the outer layer of respirator, iodinated resin incorporated on filtering layer, and TiO$_2$ coated filtering layer. This study suggested that MS2 virus decontamination efficacy of antimicrobial respirators were dependent on the antimicrobial agent and storage conditions. One should note that substituting conventional filter media of facemasks with nanofiber may reduce the airflow resistance that could lead to enhanced filtration (*Skaria & Smaldone, 2014*).

A temperature sensitive and reusable and recyclable face mask consisting of graphene-coated nonwoven filter was recently reported by *Zhong et al. (2020)*. This mask provided better protection to aqueous respiratory droplets due to its super-hydrophobic surface. Additionally, the mask can be reusable by sterilizing the surface with solar illumination. Although, viruses filtering performance of the mask has not been reported, such mask might be available as next generation face mask.

## Virus decontamination methods

Because of increased demand and subsequent shortage during viral outbreaks, surgical mask and respirator are decontaminated and reused. Reuse of these protective gears after proper decontamination may help fulfil supply chain constraints to some extent during the pandemics. However, improper decontamination and reuse of face masks and respirators may pose transmission risk. An ideal decontamination process is expected to inactivate any infectious material without altering the membrane integrity and filtering

performance. The decontamination methods can be broadly categorized as self-deactivation and forced de-activation.

In the first approach, partly discussed in earlier section, the mask material is functionalized or additional material having novel property is incorporated so as to deactivate the pathogens. One of the strategies is to functionalize fibrous filtration unit of mask by salts such as sodium chloride (*Quan et al., 2017*). In this experiment, salt coating on the fiber surface dissolved when exposed to virus aerosols. The salt destroyed the pathogens when it recrystallized during drying. The salt-coated filters also showed higher filtration efficiency than conventional mask filtration layer. The virus spiked salt treated filters provided 100% survival rate of mice. Viruses captured on salt-coated filters exhibited rapid infectivity loss compared to gradual decrease on bare filters. Salt-coated filters proved highly effective in deactivating influenza viruses regardless of subtypes and following storage in harsh environmental conditions. This simple pathogen deactivation method can be helpful in obtaining a broad-spectrum, airborne pathogen prevention device in preparation for epidemic and pandemic of respiratory diseases. Similarly, a quaternary ammonium based antimicrobial surfactant was evaluated to examine its efficiency to reduce bacterial burden on FDA cleared surgical face mask surface (*Tseng, Pan & Chang, 2016*). The antimicrobial surfactant was covalently bound onto mask surface before use. The antimicrobial mask provided >99.3% efficiency for all three bacterial species tested. Interestingly, the antimicrobial agent on the modified mask the antimicrobial agent reduced the average colony rates by 91.8% for bioaerosols that came into contact with the mask ($10^3$ CFU/m$^3$). However, the rate decreased with increased bioaerosol concentrations.

In a forced de-contamination approach, pathogen is deactivated by using external agents such as vaporized hydrogen peroxide (VGP) (*Kenney et al., 2020*), ultraviolet germicidal irradiation (UVGI), ethylene oxide (EtO), microwave oven irradiation, autoclaving (*Grinshpun, Yermakov & Khodoun, 2020*), and bleach (*Viscusi et al., 2009*). The details of such methods are described in other reviews (*Ou et al., 2020*; *Rodriguez-Martinez, Sossa-Briceño & Cortés-Luna, 2020*). Therefore, we only briefly mention some of the selected findings.

Vaporized Hydrogen peroxide (HP) can penetrate the porous fabric that may harbor virus. The virucidal activity of HP was tested in surgical N95 respirators that were aerosolized with three bacteriophages: Pseudomonas phage phi-6, T7, and T1 (*Kenney et al., 2021*). It was found that single HP vapor cycle resulted in complete eradication of the bacteriophages from the respirator.

*Viscusi et al. (2009)* compared the effectiveness of five different decontamination methods such as ethylene oxide, bleach, microwave oven irradiation, germicidal irradiation (UVGI), and vaporized hydrogen peroxide (VHP) in nine different models of NIOSH-certified respirators (surgical N95 respirators, N95 FFRs, and P100 FFRs). Each respirator was tested for five decontamination methods and the change in ordor, physical appearance, airflow resistance and aerosol penetration was studied. Also, change in material properties of the respirator and possible health risks to the user were evaluated. They found that microwave oven irradiation melted some of the samples. The scent of

bleach low levels of chlorine gas were found in the decontaminated respirators. The VHP, ethylene oxide (EtO), and UVGI, were found to be better decontamination methods.

## CONCLUSIONS AND FUTURE PERSPECTIVES

The filtering efficiency of a face mask and respirator depends on number of parameters such as nature of filter media, size of particle, and environmental parameters. The level of protection also depends on facial seal and user compliance. Several studies have shown that the N95 respirator or equivalent or higher, if worn properly, can provide expected protection to the user in high risk environments. The filtering efficiency of surgical mask is lower than the N95 respirators, and cloth face mask perform even poorer.

A few issues regarding the use of respirator and face mask are the discomfort to the user in prolonged wearing due to imperfect facial fitness, poor heat management inside the filtering device, filter clogging, and increased breathing resistance. In worst cases, these issues could even lead to psychological impacts (*Roberge et al., 2010*; *Roberge, Kim & Coca, 2012*) and reduced adherence and loss of workplace protection factor. So, there is need of next generation facemask and respirator having improved functionalities. The emerging 3D printing technology along with the development of novel materials such as metal organic framework (MOF) based filters (*Li et al., 2019*), nano-fibrous membrane containing charge enhancer (*Liu et al., 2015*), and use of material having high infrared transparency or reflectance for heat management during summer and winter seasons (*Yang et al., 2017*) will be very useful. In recent years, due to significant advancement in electrospinning technology, fabrication of filter membranes having desired fiber size, surface area, porosity, and functionality is possible. It is expected that novel facemask and respirators, which incorporate electrospun membranes, will be available commercially in future (*Cheng et al., 2017*; *Tebyetekerwa et al., 2020*; *Zhang et al., 2021*).

Currently, used respiratory protection devices pose potential risk of primary and secondary infection and transmission due to improper handling and disposal. In viral outbreaks, because of increased demand and subsequent shortage, surgical mask and respirator are decontaminated and reused. There is still a chance of infection during decontamination process or by ineffective decontamination. Also, device interiority and performance may deteriorate, and level of protection could decrease. So, there is need for better decontamination methods other than explored in *Viscusi et al. (2009)*, *Kenney et al. (2020)*. The solution to this issue could be the incorporation of filter medias that can self-decontaminate, as partly explored in *Borkow et al. (2010)*, *Fujimori et al. (2012)*, *Tseng, Pan & Chang (2016)*, or the design of a device that could incorporate a resistive heating element.

Another issue during viral outbreak, including COVID-19, is inevitable use of cloth face masks, especially in low income countries. The lower efficiency of such mask is partly due to loose facial fitting and the material used. There is a need for low-cost and effective home-made alternative fabric material to the cloth face mask. One of the possibilities for better performing cloth facemasks (*Zhao et al., 2020*) could be the use of fabrics that can be charged electrostatically so that the filtering efficiency can be increased.

### Funding

The authors received no funding for this work.

### Competing Interests

The authors declare that they have no competing interests.

### Author Contributions

- Bhanu Bhakta Neupane conceived and designed the experiments, performed the experiments, analyzed the data, prepared figures and/or tables, authored or reviewed drafts of the paper, and approved the final draft.
- Basant Giri performed the experiments, analyzed the data, authored or reviewed drafts of the paper, and approved the final draft.

### Data Availability

This is a literature review article.

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
