# Peer review of "(untitled)"

_PeerJ Materials Science, doi:10.7717/peerj-matsci.17_

## Round 0.1 · original submission · Major Revisions

In addition to addressing the reviewers' comments, the manuscript requires thorough proofreading and copyediting before publication. For example, in the abstract "...face mask and respirator minimize virus transmission by providing physical barrier to the infectious respiratory particles." should be "...facemasks and respirators minimize virus transmission by providing a physical barrier to infectious respiratory particles.", and "...prospective on the need of novel filter material..." should be "...perspective on the need for novel filter materials..." (or something similar). I recommend using a professional copyediting service or at least some free grammar-checking software (e.g., Grammarly) to help with this.

Reviewer 1 ·

Basic reporting

No comments

Experimental design

no comments

Validity of the findings

no comments

Additional comments

In this paper, the authors have presented a review of various materials and testing methods for virus filtration of face masks and respirators. The topic is hot and this article can provide in-depth details on the face masks and respirators. After careful reading, I concluded that the manuscript should be published. I suggest a minor change to the manuscript.
1) The surface area and porosity of the fibers are important features that dictate the breathability and filterability of the face masks. Electrospinning has been considered as one of the best techniques for fabricating nanofiber membranes with desired porosity and surface area. However, the electrospinning technique has been ignored in this paper. It would be better if the authors could include some examples/ or recent trends in electrospinning.
2) There are few typing errors: Line no 70, Line no 409

Reviewer 2 ·

Basic reporting

The MS is clearly written scientifically, English language is pretty much okay.

Experimental design

I think the article falls within the scope of the journal.

Validity of the findings

Abstract and conclusion are well stated.

Additional comments

Authors reviewed the virus filtering efficiency of the face masks and respirators in this MS. The manuscript is well written and presented clearly. There are some issues to be discussed more before published. The comments are as follows.
1. The importance of the face mask and respirator should be discussed in introduction part to create the background for their study.
2. The special terms such as face velocity, charge holding, etc. used in MS should be defined.
3. The proven filters/mask such as N95 are expensive to used, in this case, authors should think discussion briefly about recycling by using suitable sterilization techniques.
4. Authors should discuss how the virus penetrates the mask/respirator briefly.
5. The viruses are very much smaller than the pores of the filter materials, in this case how these materials can be able to prevent them from entering?
6. I suggest making a table mentioning filter materials (possibly polymers), their filtration efficiency, their general name such as N95, surgical face mask, etc., the application, and so on.

Reviewer 3 ·

Basic reporting

Insufficient background in the area of respiratory protection. Need to review the literature thoroughly.

Experimental design

Test methods are not well understood and are mixed up.

Validity of the findings

Conclusions address research questions.

Additional comments

Review of material and testing methods for virus filtering performance of face mask and respirator
This review describes the material and testing methods for face masks and N95 respirators, and their virus filtering performance. The difference in properties of face masks and respirators to reduce/prevent virus transmission has been described. To address respirator shortage during the COVID-19 pandemic, research on filter material with novel functionalities, new mask designs and decontamination methods have been discussed.
Aerosol particles are captured by fibrous filter materials of N95 respirators and face masks. Inert particles and virus particles are captured by the same mechanisms and literature shows there is no difference in filter efficiency. In the US, neither NIOSH nor FDA requires viral efficiency tests for approval process. The developments in this area appear to have limited applications.
The information on face masks including the approval agency and testing requirements is lacking. Facemask test methods have been mixed up with N95 respirator test methods. The distinction between filter efficiency and OSHA assigned protection factor should be well defined.
Minor comments:
Lines 66-68 – Recent studies showed COVID-19 transmission by aerosol mode. Reference shows “Organization”, which should be “World Health Organization” – change it throughout the manuscript.
Lines 66-68 – Surgical face mask. Few other names include surgical, isolation, dental, and medical procedure masks. What facemasks are described in the manuscript? US FDA cleared surgical masks (SMs) are regulated under 21 CFR 878.4040 and they meet certain efficiency for inert particles and biological particles as well as fluid barrier protection standards and standards for Class I or Class II flammability tests. The surgical mask described in the study should be identified.
Lines 89 – “surgical masks” are recommended for protection. Is this US FDA approved or other agency approved masks? Different types of masks under the name “surgical masks” are on the market. The approval standard or agency should be mentioned. FDA surgical mask information can be obtained from reference “Guidance for Industry and FDA Staff Surgical Masks - Premarket Notification [510(k)] Submissions Document issued on: March 5, 2004 and a correction posted on July 14, 2004.”
Similarly, 42CFR Part 84 explains NIOSH approval of N95 masks.

Lines 113 – In the US, respirators are certified by the NIOSH under 42 CFR Part 84.
Lines 143-158 – Filter efficiency and assigned protection factor have been described. Assigned protection factor (APF) applies to only NIOSH approved respirators not respirators for other countries. APF is not applicable to surgical masks. It appears that filter efficiency and APF have been misunderstood and simplified. Filtration efficiency and APF are not the same. One cannot simply measure the filtration efficiency and report the APF value.
The overall performance of a respirator is described by the OSHA assigned protection factor (APF). APF is defined as the workplace level of respiratory protection that a respirator or class of respirators is expected to provide to employees when the employer implements a continuing, effective respiratory protection program as specified by the 29 CFR 1910.134 standard. APF accounts for factors including filter media, faceseal, valve leakage, fit factor and others. OSHA assigns these numbers after thoroughly reviewing the available literature including the various analysis by respirator authorities, as well as quantitative analysis of data from workplace protection factor and simulated workplace protection factor studies, comments submitted to the record, and public hearing testimony. No such considerations exist for surgical masks and APF does not apply to SM.
Lines 169 – Again the test parameters are all mixed up. N95 respirator filtration test is done at 85 L/min using charge neutralized NaCl aerosol. Additional tests are needed for certification under a set of test conditions described in 42CFR Part 84. Face velocity 0.5 to 25 cm/sec is described in the ASTM standard. These two cannot be mixed up. This section should be revised.
Lines 181-183 – I assume the size of virus particles refer to physical diameters not MMAD.
In the case of N95 respirators, virus particles smaller and larger than the most penetrating particle size (MPPS) (~50 nm [physical diameter] for N95 respirators and most of the surgical masks). The virus particles are captured up to 95%.
Lines 293-303 – specify if this is a US FDA cleared surgical mask.
Lines 328-340 – For NIOSH certification, N95 respirators should meet the test requirements described in the 42 CFR Part 84 standard. There is no requirement for the number of filter media layers and the order of hydrophobic and hydrophilic layers.

Annotated reviews are not available for download in order to protect the identity of reviewers who chose to remain anonymous.

---

## Round 0.2 · Major Revisions

Several of the issues identified by one of the original reviewers have not been adequately addressed in the revised manuscript.

Reviewer 3 ·

Basic reporting

No clear understanding of the different types of masks for protection.

Experimental design

Test method for one type of mask is mixed up with another type.

Validity of the findings

Reasonable.

Additional comments

Review of material and testing methods for virus filtering performance of face mask and respirator
This revised manuscript is much improved on the description of information in several sections. Some points have been overlooked and not satisfactorily addressed.
Specific comments:
Lines 61 – (Yang et al., 2009, p. 1). What is the reason for adding p.1? just delete throughout the manuscript.
Lines 66-68 – Recent studies showed COVID-19 transmission by aerosol mode. This point has not been included in the revised manuscript.
Line 109- ‘Respirators are primarily designed for high risk environments including by health care professionals while treating a COVID patient’ – revise the sentence
Lines 112 – ‘Since respirators have special design and advanced filter materials (figure 2c), a properly worn respirator provides an excellent protection to the user’.
delete ‘excellent’, better to say ‘expected’ because N95 respirator can give 95% protection level for particulates when fit tested and properly used in workplaces with a written respiratory program in place. (A P100 respirators can give 99.97% protection level, and higher level respirators can give higher protection levels.)
Lines 117 – However, when the EV is not working properly, it may contaminate the nearby environment with infectious virus particles through exhaled breath.
Delete ‘not’. This is the reason why NIOSH recommends respirators without EV for infectious diseases. Revise the sentence.
Line 119- There are different names for respirators. Gather more information before writing this section. It is full of mistakes.
What do you mean? There are only N95, N99 and N100 in the whole world? You may want to say that there are different types of particulate respirators available in different countries. In the US, Filtering facepieces respirators (FFRs) (disposable FFRs), elastomeric half-mask, full facepiece, powered air-purifying respirators, supplied air-purifying and self-contained breathing apparatus and others are available. N, P and R series FFRs are available at 95, 99 and 99.95% efficiency levels (total 9 different types). (approval - code of federal regulations Title 42, Part 84). Then describe N95 FFRs, (widely used, reasonable cost).
Surgical N95 masks are a subset of N95 FFRs which are used in healthcare for infectious aerosols. They are NIOSH approved N95 FFRs and also cleared by FDA for fluid resistance, flammability, and biocompatibility properties. Presently, NIOSH reviews the results for fluid resistance, flammability, and biocompatibility supplied by the manufacturer for certification.


Other countries have different types of respirators. FFRs in the UK include FFP1, FFP2 and FFP3. Revise the paragraph.
Line 121- N-not resistant to oil, R-partially resistant and P-resistant (oil proof)
Line 123- The ‘medical N95’ ---------- delete. It is ‘Surgical N95’ and make the change throughout the manuscript.
Line 127- wrong reference. For certification refer; code of fed regulations. 42 CFR Part 84.
Line 131- come up with a reason why filtering efficiency for virus? filter efficiency for inert particles such as NaCl aerosol and other particles are well documented and info on virus particle filtration is not well described.??
147- may be better to use ‘---------masks and respirators ------- ‘?
159- delete ‘facial fitness, breathing condition’- not applicable for efficiency measurement.
Line 164-176- delete this part ‘Alternately------respectively’. APF does not apply to SMs. How is the APF of N95 compared with SM?
Line 180- include filter media charge and particle charge
Line 182- Delete it. ‘A user can use a filtering device in a wide range of 183 particle size and concentration distribution and breathing conditions………(Balazy et al., 2006; Ba\lazy et al., 2006; Shine, Rogers & Goldfrank, 2009, p. 1).’
Line 183- 186 -‘The regulatory recommendations for the use of respirator and surgical mask are made from in vitro measurement of filtering efficiency(Balazy et al., 2006; Ba\lazy et al., 2006; Shine, Rogers & Goldfrank, 2009, p. 1).’
Refer 42CFR Part 84 NIOSH particulate respirator standard for certification.
Line 189- ‘NIOSH respirator test method(NIOSH, 2019)’
Do not generalize test conditions because different tests are required for different types of respirators.
NIOSH uses NaCl aerosol method (not recommending) for testing only N95 FFRs, not other respirators. revise the sentence.
Line 199-200- Please delete.
No need to describe the sealing of a respirator because the test methods includes this step. Manikin system is not used for this testing.
Line 201- Make a paragraph for surgical mask testing.
Line 208-226. Delete these sentences. The NIOSH -------- Grinshpun et al 2009)
214. The US FDA also provides clearance to medical surgical mask after reviewing the
FDA clears SM after reviewing the test results and supporting information 215 provided by the manufactures in the 510(k) premarket application(US Food and Drug 216 Administration, 2004). In the 510(k) application, the manufactures are required to provide the 217 results of fluid resistance, polystyrene latex and Staphylococcus aureus bacterial aerosol filtering 218 efficiency, differential pressure, and flammability tests. Read the FDA document and include all the tests required for surgical masks. Do not confuse with surgical N95 FFRs tests.
Read the literature - Surgical N95 respirators are recommended in healthcare. These are NIOSH certified N95s and also cleared for fluid resistance, flammability and biocompatibility by NIOSH.
See reference. Rengasamy et al. 2021. American Journal of Infection Control, online 24 March 2021
Lines 218 –234 delete ‘Size of aerosol particle -----------illustrated in Fig 1. Neither NIOSH nor FDA requires this test. What is the purpose?
Line 247- The VFE of surgical masks 248 ranged from 80 to 85%, suggesting surgical masks cannot be as effective as N95 respirators for 249 small virus. Make sure it is correct.
One SM shows penetration of ~30 to 85% for ~5 to 80 nm particles while the other SM showed 5 to 20% .
Line 250- Specify the name of the SM used in the study.
Line 256 – 258 In another study, filtering efficiency of medical 257 N95 respirator was ≥99.6% for all combinations of experiment configurations using influenza A 258 virus, rhinovirus 14, and bacteriophage ΦΧ174 at a flow rate of 28.3 Lmin-1(Zhou et al., 2018).
There is no medical N95. The name of the mask is ‘Surgical N95’
Line 309-321- smaller fiber, longer fiber, shorter fiber, - use smaller and larger with diameter
Line 511- excellent protection. delete ‘excellent’ and use expected as suggested previously.

---

## Round 0.3 · accepted · Accept

I am now satisfied with the revisions made to the manuscript.

Reviewer 1 ·

Basic reporting

No comment.

Experimental design

No comment.

Validity of the findings

No comment.

Additional comments

The quality of the manuscript has been improved after revision.

Reviewer 2 ·

Basic reporting

The authors revised the manuscript substantially.

Experimental design

The study is focussed on the subject they choose.

Validity of the findings

Conclusions are supported by results

Additional comments

The revision is satisfactory.